# Hidden paths to endless forms most wonderful: ecology latently shapes evolution of multicellular development in predatory bacteria

Marco La Fortezza [1✉], Olaya Rendueles[1,2], Heike Keller[1] & Gregory J. Velicer [1]

Ecological causes of developmental evolution, for example from predation, remain much investigated, but the potential importance of latent phenotypes in eco-evo-devo has received little attention. Using the predatory bacterium *Myxococcus xanthus*, which undergoes aggregative fruiting body development upon starvation, we tested whether adaptation to distinct growth environments that do not induce development latently alters developmental phenotypes under starvation conditions that do induce development. In an evolution experiment named MyxoEE-3, growing *M. xanthus* populations swarmed across agar surfaces while adapting to conditions varying at factors such as surface stiffness or prey identity. Such ecological variation during growth was found to greatly impact the latent evolution of development, including fruiting body morphology, the degree of morphological trait correlation, reaction norms, degrees of developmental plasticity and stochastic diversification. For example, some prey environments promoted retention of developmental proficiency whereas others led to its systematic loss. Our results have implications for understanding evolutionary interactions among predation, development and motility in myxobacterial life cycles, and, more broadly, how ecology can profoundly shape the evolution of developmental systems latently rather than by direct selection on developmental features.

[1] Institute for Integrative Biology, ETH Zurich, Universitätstrasse 16, 8092 Zurich, Switzerland. [2] Microbial Evolutionary Genomics, Institut Pasteur, CNRS, UMR3525, 75015 Paris, France. ✉email: marco.lafortezza@env.ethz.ch

Ecological context fundamentally shapes the remarkable diversification[1] of developmental programs, a notion that precedes the modern field of evolutionary developmental biology, or evo-devo[2,3]. A recently-emerged focus on eco-evo-devo seeks to quantitatively understand such ecological causation of developmental evolution[4–6]. Many biotic ecological factors, such as predator–prey[7–9] and social interactions[10–12], as well as abiotic factors such as temperature[13–15] and oxygen level[16–18], are hypothesized or known to play important roles in shaping the evolution of developmental features[5,19], including developmental plasticity[20–26]. Direct tests for such ecological causality can be performed with experimental evo-devo, i.e., evolution experiments that examine hypotheses about multicellular development[27].

Many eco-evo-devo studies focus on direct selective relationships, i.e., on how a developmental process responds evolutionarily to selection by an ecological factor on that process. However, the genetic basis of a phenotype that becomes important to fitness at some stage of a lineage's history might have first evolved in a hidden, non-adaptive manner. Many phenotypes are manifested in only a limited set of ecological contexts[28], which have been referred to as "inductive environments"[29,30] (see Methods: Semantics). For example, in animals, exposure to novel dietary conditions or unusual temperatures can induce the expression of new morphological phenotypes that would otherwise stay hidden[31,32]. The genetic basis of a focal phenotype might first evolve in a non-inductive environment (by any evolutionary mechanism, e.g. adaptation or genetic drift), while the actual phenotype is only first manifested and thus potentially exposed to selection after later exposure to an inductive environment. Such evolution of initially latent phenotypes — latent-phenotype evolution (LPE[33,34], see Methods: Semantics—is common[33–37], contributes to evolutionary diversification[33,34] and can fuel some forms of exaptation[28,38–40].

Studying the evolutionary history of developmental-phenotype causation is notoriously challenging in most natural systems, for which the relevant details of genetic, phenotypic, and ecological history are often unknown. This is perhaps particularly true for organisms for which multicellular development is obligate and for deciphering whether the genetic basis of a focal phenotype first evolved under environmental conditions in which that phenotype was not manifested. Ideal organisms for such studies would be both amenable to experimental evolution and characterized by a developmental process that is facultative and environment specific and which generates readily-quantifiable morphological phenotypes.

Microbes that undergo genetically-programmed multicellular development in response to specific environmental conditions are powerful model systems for experimental studies in eco-evo-devo[41]. Among prokaryotes, predatory myxobacteria are probably the most recognized in this regard, especially the model species *Myxococcus xanthus*[41], which is found in many soil habitats worldwide[42]. In the vegetative-growth phase of their multicellular life cycles, many myxobacteria kill and consume diverse microbes as prey (both prokaryotic and eukaryotic) by molecular mechanisms that remain to be well-characterized[43,44]. *M. xanthus* employs two mechanistically distinct (but pleiotropically connected) motility systems that are each regulated by many genes[45] and together allow cells to migrate in search of prey and other resources and to carry out aggregative multicellular development. One system, traditionally known as "S-motility," is mediated by Type-IV pili[46], and the other, traditionally known as "A-motility," is hypothesized to function by active transport of focal-adhesion protein complexes attached to the substratum[47]. The developmental phase of myxobacterial life cycles exhibited by many species is triggered by nutritional depletion, upon which cells aggregate to cooperatively construct elevated multicellular fruiting bodies. Some cells within fruiting bodies differentiate into spores capable of surviving harsh environmental conditions such as extended nutrient deprivation, high temperatures, and likely predation[45,48,49].

Pleiotropic effects of adaptation—whether exerted across ecologically distinct growth environments[50–52] or across distinct life-history stages[41,53–58]—are, collectively, a central feature of evolution. Focusing on distinct life-history stages, degrees of pleiotropic connectedness among the genetic systems that underlie stage-specific traits are critical determinants of how life cycles can evolve[58,59]. Genetic and evolutionary connections between growth vs developmental phases of myxobacterial life cycles have received some attention[60–64] but require much further investigation. Adaptation to some prey environments fueling population growth may pleiotropically impact fruiting body formation during subsequent bouts of starvation. Indeed, the adaptation of *M. xanthus* to predatory growth while foraging for *Escherichia coli* in a previous study resulted in a general decrease in fruiting body numbers during starvation, suggestive of a tradeoff between predatory fitness and development[61]. Trade-offs from systematically antagonistic pleiotropic effects of mutations adaptive in one life-history phase on performance at another phase[54,65,66] have major implications for life-cycle evolution as a whole and might promote some forms of diversity[66,67]. However, the occurrence and character of such life-cycle pleiotropy may often be specific to the ecological context in which the causal mutations evolve[66], a hypothesis we test here.

Fruiting body morphology has diversified strikingly among myxobacterial species (order Myxococcales)[41], as have many developmental traits already at the intra-specific level among *M. xanthus* lineages, including fruiting body traits[11,68,69], nutrient[70] and cell-density[71] thresholds triggering development, developmental timing[70] and even the genetic elements required for development[72]. However, the relative contributions of various forms of selection and stochastic forces to developmental diversification among myxobacteria remain unclear[41].

Forms of selection that may play major roles in diversifying fruiting body morphologies include (i) selection by abiotic or biotic ecological factors on morphological features per se, (ii) selection on non-morphological aspects of development (e.g., the timing of cell-cell signaling) that may nonetheless impact final fruiting body morphology and (iii) selection on non-developmental life-cycle behaviors such as predation and growth-phase motility that pleiotropically impact development. Different biotic and abiotic factors such as nutrient level[73], substrate stiffness[74], and social environment[60,69,75] are known to affect *M. xanthus* developmental phenotypes, with likely implications for the evolution of fruiting body morphology[11]. The experimental inducibility and facultative character of *M. xanthus* multicellular development and the versatility for experimental evolution make this bacterium a powerful system for investigating how developmental phenotypes can evolve latently (see definition of "latent phenotype" in Methods: Semantics). Such phenotypes include the character and degree of phenotypic plasticity, which can be important for fitness and future evolution across variable environments[29,74,76,77].

We test the hypothesis that the ecological context of evolution by *M. xanthus* during vegetative growth can determine how such evolution latently alters developmental phenotypes expressed only upon starvation. To do so, we use evolved populations from MyxoEE-3 (Fig. 1a)[78], an evolution experiment in which *M. xanthus* populations were selected for increased fitness at the leading edge of growing colonies that swarmed across agar-surface environments that differed at one or more ecological variables (Fig. 1b). Importantly, MyxoEE-3 populations were not experimentally subjected to starvation-induced development,

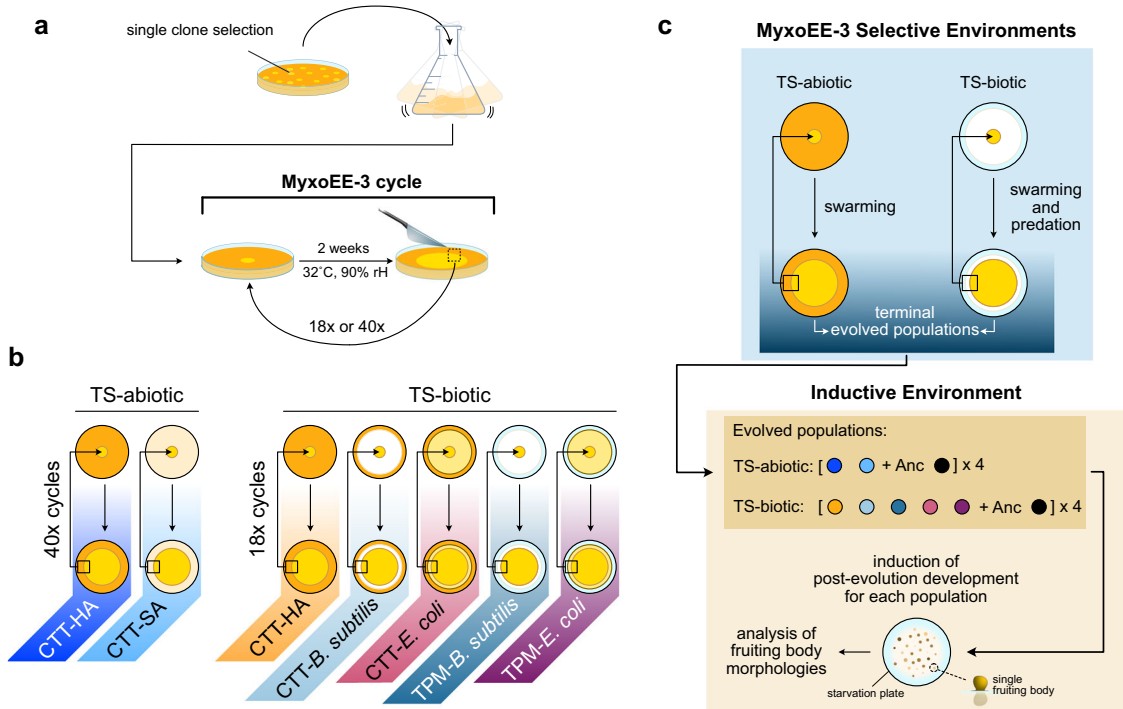

**Fig. 1 Overview of MyxoEE-3 treatments and time-points examined here for latent evolution of fruiting body morphology. a** MyxoEE-3 initiation and cycling scheme (adapted from ref. [104]). Ancestral subclones (collectively 'Anc') were isolated, grown in liquid, and used to start MyxoEE-3 populations (see Supp. Table 1). During each evolutionary cycle, populations (yellow circles) were allowed to swarm for 2 weeks at 32 °C and 90% rH before a small rectangle at the furthest point from the swarm center was cut and transferred upside down to the center of a new plate. Populations examined here underwent either 18 or 40 such cycles according to the specific treatment set (TS). **b** Overview of MyoxEE-3 treatments relevant to this study subdivided into two treatment sets (TS)—abiotic and biotic. TS-abiotic populations (yellow circles) evolved on either CTT hard agar (CTT-HA, dark orange plates) or CTT soft agar (CTT-SA, beige plates) for a total of 40 cycles. TS-biotic populations evolved on either CTT-HA only (same as CTT-HA in TS-abiotic) or on CTT-HA (dark orange plates) or TPM-HA (light blue plates) with lawns of either *Bacillus subtilis* (white area) or *Escherichia coli* (light yellow area). The TS-biotic populations were examined after 18 MyxoEE-3 cycles. **c** Scheme representing the distinctive experimental selective and inductive environments that characterize this study. Populations evolved in a range of different environments (selective environments) (panel **b**), where the abiotic (TS-abiotic) and biotic (TS-biotic) sources of selection acted on motility and/or predation during vegetative growth. Importantly, while exposed to selection during MyxoEE-3, transferred populations did not undergo fruiting body formation, such that development per se was hidden from the selection. Post-evolution development was subsequently induced on starvation plates (inductive environment) in all MyxoEE-3 terminal populations independently to study the latent evolution of fruiting body morphology.

such that evolutionary change in development-specific traits does not reflect adaptation for improved developmental fitness, but rather either pleiotropic effects of selection on other traits or stochastic evolution (Fig. 1c).

To test how different ecological contexts during vegetative growth can affect the latent evolution of development, we analyzed two distinct sets of MyxoEE-3 treatments (Fig. 1b and Supp. Table 1)[79]. In one treatment set (TS), populations evolved in selective environments designed to differ only abiotically (TS-abiotic) (Fig. 1b, c). Specifically, replicate populations evolved on either of two high-nutrient agar surfaces, one stiff surface with a high agar concentration (1.5%, CTT hard agar, or "CTT-HA") and one soft surface with a low agar concentration (0.5%, CTT soft agar, or "CTT-SA"). The TS-abiotic treatments were analyzed after 40 cycles of MyxoEE-3 selection. On 0.5% agar, swarming of *M. xanthus* reference strains is driven almost exclusively by Type-IV-pili-mediated S-motility, whereas swarming on 1.5% agar is driven by a combination of A-motility and S-motility[61,78,80].

The second set of treatments (TS-biotic) included populations from five MyxoEE-3 treatments (Supp. Tables 1, 2), including one shared with TS-abiotic (CTT-HA) and four treatments in which *M. xanthus* was offered either *Bacillus subtilis* or *E. coli* bacterial species (Gram + and Gram−, respectively) as prey (Fig. 1b, c). Among the four TS-biotic treatments with prey, two were

identical to the CTT-HA treatment, except that a lawn of either *B. subtilis* (*B. subtilis*-CTT) or *E. coli* (*E. coli*-CTT) was allowed to grow on the CTT hard-agar culture plates prior to their inoculation with *M. xanthus*. In these treatments, both the prey and any nutrients unused by the prey that remained in the agar substrate were available to *M. xanthus*. In the two other TS-biotic treatments (*B. subtilis*-TPM and *E. coli*-TPM), prey were first grown to stationary phase in high-nutrient CTT liquid and then spread as high-density lawns onto buffered TPM agar lacking casitone, the carbon growth substrate present in CTT, before plates were inoculated with *M. xanthus*. In these treatments, nutrients were only available to *M. xanthus* from the prey. All TS-biotic treatments had plates with high (1.5%) agar concentration and were analyzed after 18 cycles of MyxoEE-3 (Fig. 1b).

Among the evolved TS-abiotic and TS-biotic populations of MyxoEE-3 examined here and their ancestors (Supp. Table 1, see *Methods: MyxoEE-3*), we analyzed variation in four morphological traits associated with fruiting body formation as well as variation at spore production (Fig. 1c). We quantified the degree of phenotypic divergence from the ancestor and tested for effects of MyxoEE-3 selective environments on developmental LPE, including for treatment-level means for each trait, phenotypic plasticity across an environmental gradient for the TS-abiotic treatments and degrees of stochastic diversification.

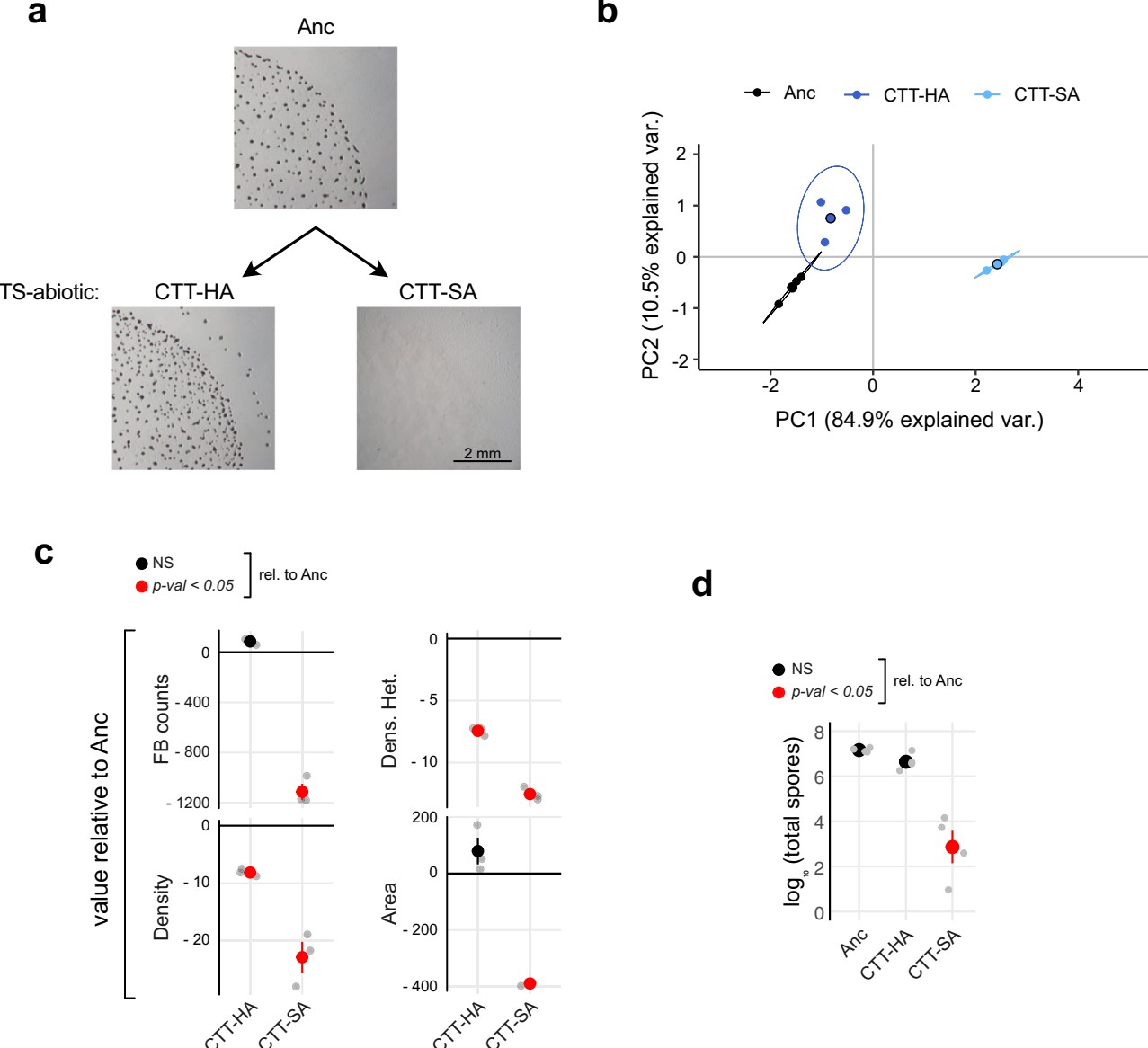

**Fig. 2 Surface stiffness shapes developmental LPE. a** Representative developmental phenotypes of the ancestor (Anc) and TS-abiotic evolved populations from CTT-HA (Population 3, P3; see Supp. Table 1) and CTT-SA (P39) (scale bar = 2 mm). **b** PCA based on four morphological traits showing the overall treatment-level phenotypic differentiation among the CTT-HA (dark blue) and CTT-SA (light blue) TS-abiotic treatments (MyxoEE-3 cycle 40) and their ancestor (black). Large circles represent average morphospace localization (centroids) obtained from three independent biological replicates (small circles, $n = 3$). Ellipses represent 95% confidence regions, while percentage values on the x and y axis report the variation explained by the principal components PC1 and PC2, respectively (Supp. Fig. 1A). **c** Mean values ± SEM of each analyzed morphological trait relative to the ancestral levels (Anc, black-horizontal line in the graphs) ($n = 3$). **d** Mean values ± SEM of $\log_{10}$-transformed spore counts ($n = 4$). In both **c** and **d**, red and black circles indicate significant ($p < 0.05$) and non-significant (NS) differences from Anc levels, respectively. (Significance was estimated for both **c** and **d** with one-way ANOVA followed by two-tailed Tukey tests. $p$ values of all comparisons of evolved treatments with Anc, as well as all pairwise comparisons between evolved treatments, are reported in Data Fig. 2).

## Results

**Substrate stiffness determines the character of developmental LPE.** We quantitatively compared the LPE of fruiting body morphology across MyxoEE-3 populations from both the TS-abiotic and TS-biotic treatment sets and the ancestor (Anc). To do so, we used previously published methods of quantifying four morphological fruiting body traits[11]: three trait medians measured at the resolution of individual fruiting bodies (density, density heterogeneity, and area) and total fruiting body number per assay plate (see Methods: Induction of development and Image acquisition and trait quantification; also ref. [11]). Considering TS-abiotic

first, microscopic observation (Fig. 2a), PCA of the entire morphological-trait data set (Fig. 2b) and individual-trait analysis (Fig. 2c, d) collectively indicate that both TS-abiotic treatments diverged morphologically from the ancestor but did so in a treatment-specific manner. Indeed, perMANOVA of PCA values demonstrates morphological differentiation between the lines evolved on soft agar (CTT-SA) vs on hard agar (CTT-HA) (perMANOVA: $F = 42.9$, R2 = 0.94, $P = 0.005$).

PC1 explained 85% of the total variance (Supp. Fig. 1a), with all analyzed traits making similar contributions to shaping the fruiting body morphospace and showing strong correlations with

each other, as reflected by the similar positioning in the multivariate space in Supp. Fig. 1b. Strong overall trait correlations reflect high levels of morphological integration[59,81]. We asked whether trait correlations might evolve differently in different ecological contexts by calculating a measure of morphological integration for both TS-abiotic treatments individually (see Methods: Morphological analysis[82,83]. This analysis suggests that trait relationships changed little during MyxoEE-3 evolution on soft agar (Supp. Fig. 1c), whereas trait integration appears to have decreased moderately on hard agar. Thus, soft vs hard agar may have differentially impacted how trait relationships evolve.

The PCA suggested that evolution on soft agar led to greater morphological divergence from Anc by the CTT-SA populations than did evolution on hard agar (Fig. 2b) by the CTT-HA populations. In further support of this observation, k-means and hierarchical-clustering analyses based on the degree of morphological similarity in the morphospace also found the CTT-SA populations to be collectively distinct from both the CTT-HA populations and Anc, which together formed one statistical cluster (Supp. Fig. 1d–f).

To understand how the focal morphological traits evolved in detail and whether the distinct MyxoEE-3 environments impacted these traits differently, we also analyzed them individually. Average fruiting body density and heterogeneity decreased on average from Anc among both CTT-HA and CTT-SA populations, whereas only CTT-SA populations evolved greatly from Anc in fruiting body area and total counts, doing so by largely losing the ability to form fruiting bodies at all (Fig. 2c and Supp. Fig. 1g). CTT-HA populations remained very similar to Anc in the latter two traits, thereby explaining their lower overall divergence from Anc across all traits. We then also examined LPE of sporulation efficiency and found trends comparable to those observed for the overall morphological analysis, in that CTT-HA populations retained levels of sporulation similar to Anc, whereas spore counts drastically decreased in CTT-SA populations (Fig. 2d). Thus, under high-nutrient conditions (CTT medium), adaptation to swarming on a soft surface traded off against proficiency at both fruiting body formation and sporulation, whereas adaptation to swarming across a harder surface did not.

Considering the systematic latent loss of development in the CTT-SA lines, we examined previously reported mutation profiles of clones isolated from each of the CTT-HA and CTT-SA cycle-40 populations for possible candidate mutations[79] (one clone per population). In neither treatment were mutations preferentially localized among genes reported to be transcribed in a development-specific manner[79,84] (Supp. Fig. 2 and Supp. Table 3), as might be expected given the lack of selection on development during MyxoEE-3. However, an instance of gene-level parallel mutation specific to the CTT-SA treatment may represent the most common genetic route of developmental degradation among the CTT-SA populations. Among all loci mutated in more than one population (whether in CTT-HA or CTT-SA), the gene *lonD* was a hotspot of selection uniquely in the CTT-SA treatment, in which it was mutated in four populations (P) (P29, P31, P33, and P35)[79], whereas it was not mutated in any CTT-HA population. *lonD* (aka *bsgA*) encodes an ATP-dependent protease that is required for developmental proficiency[85,86]. The histidine-kinase gene *MXAN_5852* (*MXAN_RS28370*) was also mutated in parallel in the CTT-SA treatment (four populations). However, previous experimental mutation of this gene did not cause major decreases in spore production[87], suggesting that the MyxoEE-3 mutations found in this gene may not be responsible for the losses of developmental proficiency observed in the respective populations. We thus

speculate that mutation of *lonD* is likely to have been a shared route that led to the loss of fruiting body development among four CTT-SA populations, while at least one other route was taken by the other two CTT-SA populations examined here.

While all six TS-abiotic CTT-HA populations retained relatively high sporulation levels compared to the CTT-SA populations, three nonetheless evolved partial reductions of spore production (Supp. Table 3). The sequenced clones from those three populations each had a mutation in the histidine-kinase gene *hsfB*, which was not mutated in the other three populations that retained full ancestral proficiency at development. HsfB has been shown to phosphorylate the response regulator HsfA, which in turn activates the transcription of *lonD*[88]. Thus, gene members of the same *hsfB/hsfA/lonD* developmental regulatory pathway were clearly targeted by selection in both the CTT-HA and CTT-SA treatments, but the specific gene targeted differed between treatments (see Supp. Table 4 for all found mutations).

**Latent evolution of developmental plasticity.** Beyond phenotypes manifested in a single environment, the degree and character of developmental plasticity across environmental gradients might also evolve latently and thus, at least initially, remain hidden from selection[89,90]. *M. xanthus* developmental phenotypes often vary as a function of agar concentration[74], for example as is observed for the MyxoEE-3 ancestor at all four focal developmental traits examined here (Fig. 3a). We tested whether developmental-phenotype reaction norms of TS-abiotic populations across an agar-concentration gradient (0.5, 1.0, and 1.5%) evolved during MyxoEE-3 and whether MyxoEE-3 selective environments impacted the character of such latent reaction-norm evolution (Supp. Fig. 3a, b).

The reaction norms of both CTT-HA and CTT-SA populations evolved latently at the treatment level (Fig. 3a and Supp. Fig. 3c–e) but in very different manners. In the CTT-HA treatment, average reaction norms were altered significantly relative to the Anc clones in overall mean, linear slope, and/or shape for three of the four examined traits (Fig. 3a). Intriguingly, however, considering average reaction norms at the treatment level masks extensive stochastic diversification of reaction norms across individual populations at all four morphological traits (Fig. 3b). Because replicate populations within each treatment adapted to the same environmental conditions, diversification among them cannot be explained by systematic selective differences, but only by stochastic variation in the identity, timing, order, and the number of mutations that occurred among the replicate populations[79,91,92]. For all four traits, some replicate populations varied not only in the slope but in the very sign of the trait-value vs. agar-concentration relationship, whether across all three agar concentrations or only two (Fig. 3b). Thus, for this treatment, although reaction norms did evolve mildly at the treatment level (Fig. 3a), stochastic variation in mutational input between replicate populations mattered much more for reaction-norm evolution than did the selective conditions of the CTT-HA environment of MyxoEE-3. Intriguingly, this stochastic radiation led to repeated examples of increased plasticity (e.g., P3 and P11 for fruiting body counts), phenotypic canalization (reaction-norm slope reduction, e.g., P9 and all CTT-SA populations for fruiting body counts), qualitatively novel reaction-norm patterns (e.g., P9 for fruiting body counts) and evolutionary reversals of reaction-norm slopes (e.g. P5 and P9 for fruiting body area) (Fig. 3b).

In contrast to the CTT-HA treatment, all six of the CTT-SA populations evolved in parallel to become completely or largely non-plastic. This is because the previously noted inability of these populations to form fruiting bodies on 1.5% agar (Fig. 1c) was also observed at the other two agar concentrations (Fig. 3a and

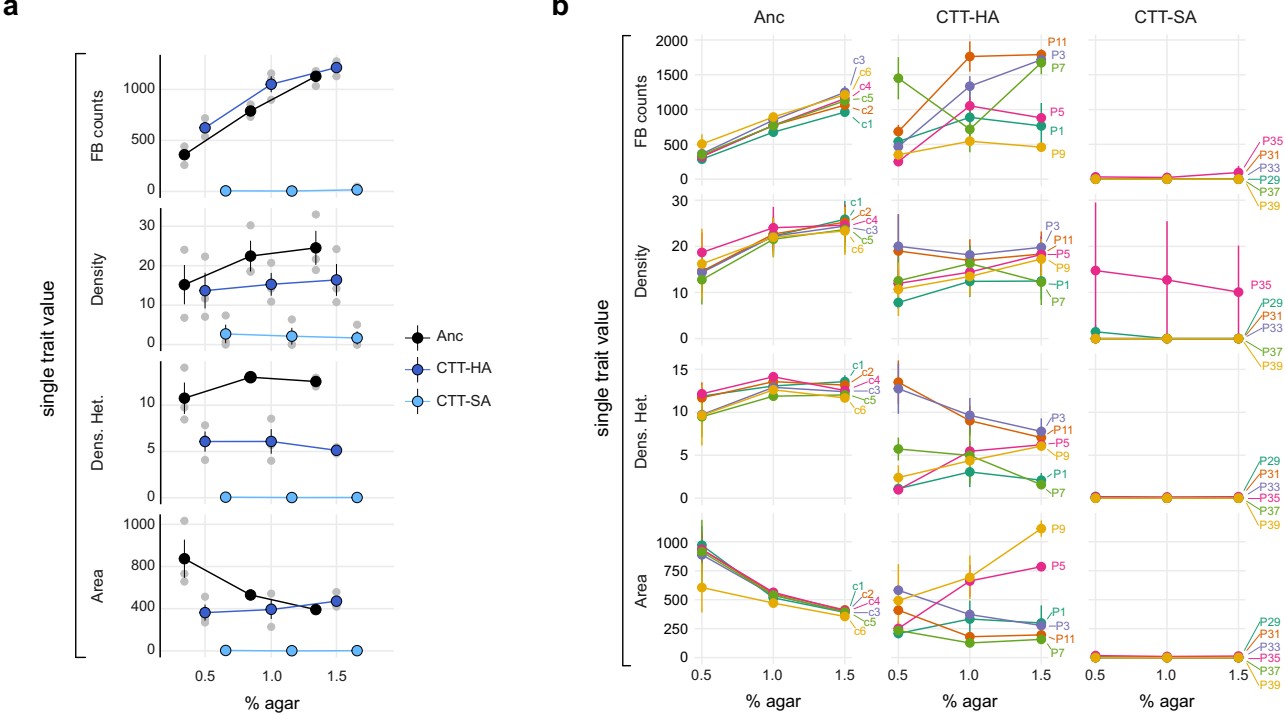

**Fig. 3 Deterministic and stochastic evolution of latent reaction norms.** Reaction norms of four morphological traits across three agar concentrations (0.5, 1.0, and 1.5%, x-axis in both panels). **a** Reaction norms across ancestral clones (black circles and lines) and treatment-level average trait values (CTT-HA and CTT-SA evolved populations, dark and light blue circles and lines, respectively). $n = 3$ independent replicates (gray circles). **b** Reaction norms for each individual Anc sub-clone (c1–c6) and each TS-abiotic evolved population across the three biological replicates ($n = 3$). Labels at the end of each reaction norm indicate the identity of the ancestral clone or evolved population. In both **a** and **b**, large circles and error bars indicate the mean ± SEM, respectively. (Significance of variables' contribution and their interaction were calculated for both **a** and **b** with two-way ANOVA. Employed models and results of all comparisons are reported in Data Fig. 3).

Supp. Fig. 3b). Also, unlike CTT-HA, which allowed much stochastic latent diversification of reaction norms, CTT-SA greatly constrained reaction-norm LPE to yield nearly identically flat evolved reaction norms across populations (Fig. 3b).

**Prey presence and identity also strongly shape developmental LPE.** Myxobacterial adaptation to distinct prey environments might drive diversification of many traits, including predatory performance and mechanisms across diverse prey types[93], social competitiveness[64,94], forms and degrees of cooperation during predation[95], motility performance[33,61], secondary-metabolite production, production of or sensitivity to antibiotics and, a hypothesis we test here, development-related phenotypes. Overall, PCA and perMANOVA of the entire morphological datasets run on all five TS-biotic treatments, four of which included prey, showed a clear evolutionary change in fruiting body morphology relative to Anc and morphological divergence among treatments (Fig. 4a, b) (perMANOVA: $F = 50.5$, R2 $= 0.95$, $P = 0.001$).

As for the TS-abiotic analysis, PCA of the entire TS-biotic population data set revealed that the primary component explains a large majority of the variance (93%) (Supp. Fig. 4a) and high overall levels of trait correlation, as reflected by the similar positioning of the four traits in the morphospace (Supp. Fig. 4b). However, we again tested whether the character of trait relationships may have diverged as a function of MyxoEE-3 selective environments by estimating their morphological integration (see Methods: Morphological analysis). Overall trait correlation appears to have changed very little relative to Anc among the *B. subtilis*-CTT populations and only mildly among the CTT-HA and *B. subtilis*-TPM populations (Supp. Fig. 4c). However, trait

correlations appear to have changed much more in the two *E. coli* treatments, decreasing relative to Anc and the other treatments. These variable outcomes suggest that different forms of selection imposed by distinct MyxoEE-3 ecological contexts—including prey identity—can differentially impact trait correlations and thus likely the degree of pleiotropic connectedness among traits[59].

Most pairwise treatment comparisons reveal treatment-level diversification at one or more developmental traits (Fig. 4c and Supp. Fig. 4g). Most broadly, the presence of prey, irrespective of abiotic context, tended to promote greater evolutionary change, as the four treatments that included prey bacteria all clearly or apparently diverged more from Anc than did the cycle-18 CTT-HA populations (Fig. 4b, c), which, like the cycle-40 CTT-HA populations examined in TS-abiotic (Fig. 2b, c), remained relatively similar to Anc.

Three TS-biotic treatments shared the same abiotic environment of high-nutrient CTT hard agar, differing only in the presence or absence of any prey or in prey identity—*B. subtilis* or *E. coli* (CTT-HA, *B. subtilis*-CTT, *E. coli*-CTT). Among these three abiotically identical treatments, the presence of either prey species resulted in greater morphological evolution than did the absence of prey (Fig. 4b, c). However, the two prey species had different effects on LPE, with greater indirect morphological evolution occurring in the *B. subtilis*-CTT treatment than in the *E. coli*-CTT treatment (Fig. 4b, c).

Consistent with this result, *B. subtilis* also caused greater LPE than *E. coli* between the *B. subtilis*-TPM vs *E. coli*-TPM treatments (Fig. 4b, c). Indeed, prey identity mattered even more for LPE than the differences between the CTT vs TPM treatments with prey (Supp. Table 2), in that both *B. subtilis* treatments showed greater divergence from Anc and from the CTT-HA

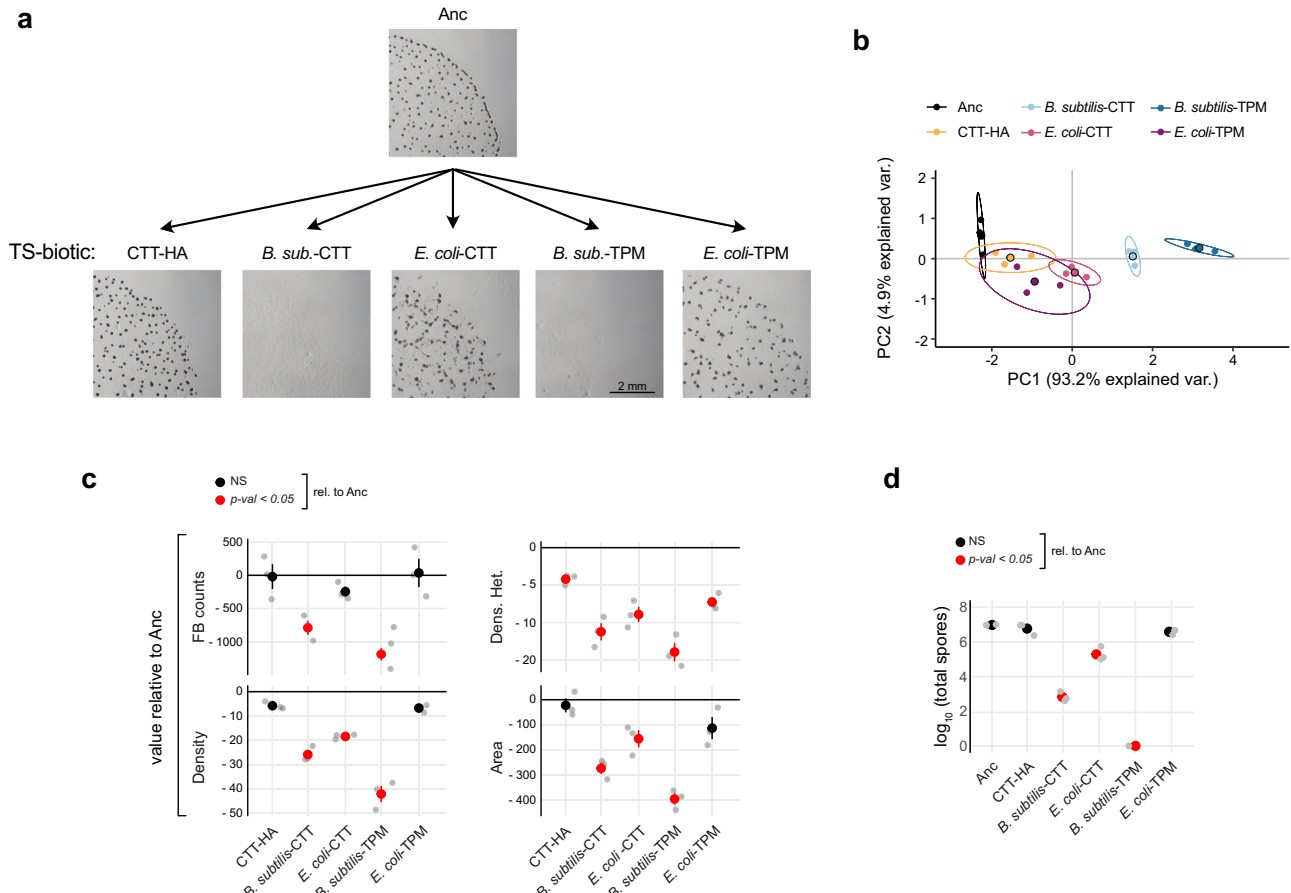

**Fig. 4 Prey presence and identity shape developmental LPE. a** Developmental phenotypes of representative TS-biotic evolved populations from each selective environment and Anc (CTT-HA = P1; *B. subtilis*-CTT = P99; *E. coli*-CTT = P93; *B. subtilis*-TPM = P133; *E. coli*-TPM = P127). **b** PCA of overall morphological divergence across all five TS-B evolutionary treatments (MyxoEE-3 cycle 18) and the Anc subclones. Large circles represent average morphospace localization (centroids) obtained from three independent biological replicates (small circles, $n = 3$), while ellipses represent a 95% confidence region. Percentage values on the x and y axis report the variation explained by the two principal components, PC1 and PC2, respectively (Supp. Fig. 5B). **c** Mean values ± SEM of individual developmental traits relative to the ancestral (Anc) levels (black-horizontal line in each graph) ($n = 3$). **d** Mean values ± SEM of $log_{10}$-transformed spore counts obtained after five days of starvation ($n = 3$). In both **c** and **d**, red and black circles indicate significant ($p < 0.05$) and non-significant (NS) differences from Anc levels, respectively. (Significance was calculated for both **c** and **d** with one-way ANOVA followed by two-tailed Tukey tests. $p$ values of all comparisons of evolved treatments with Anc, as well as all pairwise comparisons between evolved treatments, are reported in Data Fig. 4).

treatment than either *E. coli* treatment (Fig. 4b, c). Highlighting this primary effect of prey identity, k-means cluster analysis of all five TS-biotic treatments identified two primary statistical clusters distinguished solely by the presence or absence of *B. subtilis* (Supp. Fig. 4d–f). Consistent with the above collective analyses (PCA and k-means), both *B. subtilis* treatments exhibited significantly reduced trait values for all four individually examined morphological traits, whereas the *E. coli* treatments each diverged from Anc only at a subset of traits (Fig. 4c).

As for TS-abiotic, we examined levels of sporulation from all TS-biotic evolved populations. Also in this case, MyxoEE-3 selective environments strongly determined the LPE of sporulation. In detail, sporulation level decreased greatly relative to Anc for those populations evolved in the presence of *B. subtilis* in either abiotic context and in the *E. coli*-CTT populations, whereas it remained near the ancestral level for both CTT-HA and *E. coli*-TPM evolved populations (Fig. 4d).

In *Myxococcus* literature, sporulation is frequently, if not always, associated positively with fruiting body formation[41,96]. While the two developmental processes are indeed often linked, we have recently shown that sporulation can evolutionarily become decoupled from fruiting body development[11]. We

observed that MyxoEE-3 treatments that evolved large reductions in morphological-trait values also showed large decreases in sporulation (Fig. 4c, d) and therefore formally tested whether sporulation and fruiting body counts correlate positively after MyxoEE-3 evolution (for all TS-abiotic and TS-biotic treatments pooled together). Indeed, reductions in fruiting body counts were quantitatively associated with reduced spore counts ($r = 0.88$, $p = 4.4 \times 10^{-15}$) (Supp. Fig. 5).

Taken together, our results indicate that both abiotic and biotic details of selective environments during myxobacterial growth can greatly impact the evolution of latent developmental phenotypes that are revealed only upon exposure to starvation. Specifically, agar concentration in the physical substrate on which *M. xanthus* evolved and the identity of prey consumed by *M. xanthus* both had large indirect effects on the evolutionary fate of fruiting body morphology and sporulation efficiency.

**Different degrees of developmental LPE are not explained by mere differences in genome-evolution rates.** The clearly different effects of MyxoEE-3 environments on the degree of developmental LPE (Figs. 2–4) might be explained by several distinct

hypotheses. A first major hypothesis is that distinct MyxoEE-3 environments may have caused systematic differences in the temporal rate of genomic evolution across treatments, i.e., differences in the average number of mutations that accumulated in evolved populations over the same duration of MyxoEE-3. Any such differences in the rate of genomic evolution might, in turn, be due to differences in selection strength between treatments and/or to differences in mutation supply caused by differences in the average degree of per-cycle population growth across treatments. A second hypothesis is that distinct sets of mutations selectively favored by the various MyxoEE-3 treatments differ, on average, in their effects on developmental phenotypes.

We used two approaches to evaluate these hypotheses. For the TS-abiotic treatments, we examined published sequencing results (Supp. Table 4 of ref. [79]) to find that clones isolated from the CTT-HA vs CTT-SA populations did not differ significantly in the number of mutations that differentiated them from their clonal ancestor after 40 cycles of MyxoEE-3 evolution (11.7 vs. 14.4 mutations on average, respectively; $p = 0.068$ for a treatment-level difference, Welch two-samples $t$-test; df = 20.98, $t = -1.922$; with one mutator clone from P29 excluded from analysis). Thus, mere differences in the average number of mutations accrued during MyxoEE-3 do not appear to explain the vast differences in the degree of developmental-phenotype evolution undergone by the CTT-HA and CTT-SA treatments (Fig. 2).

We currently lack sequence data for the TS-biotic treatments after 18 cycles. Therefore, instead tested whether the large differences in developmental-phenotype evolution undergone by these treatments (Fig. 4) might potentially be correlated with either (i) average swarm size (considered as a proxy of population size) achieved within each 2-week MyxoEE-3 cycle of swarming and growth or (ii) differences in the average degree of swarming-rate evolution undergone by respective population sets in their MyxoEE-3 selective environment. To quantify these parameters, we performed swarming-rate assays for both the ancestors and all TS-biotic evolved populations in the respective MyxoEE-3 selective environments of the latter (Supp. Fig. 6a). We find that the degree of treatment-level morphological distance between evolved populations and Anc (as quantified by PCA) does not correlate with either (i) MyxoEE-3 selective-environment swarming rates, whether for ancestral clones or evolved populations (Supp. Fig. 6a, c) or (ii) the degree of evolutionary change in selective-environment swarming rates (Supp. Fig. 6b, d; $r = 0.29$, $p = 0.63$). Had correlations been observed, they might have reflected differences in mutation numbers accumulated during MyxoEE-3. In the absence of such correlations and in light of the mutation-number analysis of the TS-abiotic populations, the second hypothesis above is much more plausible than the first. It appears that distinct mutation sets that evolved across treatments have different effects on developmental phenotypes, on average, as a function of ecological differences between MyxoEE-3 environments.

**Distinct selective environments differentially limit stochastic diversification of fruiting body morphology.** Replicate experimental populations evolving under the same selective pressures often diversify stochastically[97,98]. However, the character of the selective environment can limit the degree of such stochastic diversification, whether for immediately manifested or latent phenotypes[11,33,34,99]. For example, we have recently shown that social selection acting during *M. xanthus* development can limit the degree of stochastic morphological diversification among evolving populations[11]. Here we asked whether such limitation of stochastic developmental diversification can also occur when

selection operates solely during *M. xanthus* population growth rather than on starvation-induced development. To do so, we quantified morphological dispersion across all replicate populations within each individual treatment of both TS-abiotic and TS-biotic from the previously obtained morphospaces (Figs. 2b, 4b) (see Methods: Morphological analysis).

We find that both agar concentration in TS-abiotic and the presence of prey (of either identity) in TS-biotic determined the degree of stochastic evolutionary diversification at developmental phenotypes during MyxoEE-3. In TS-abiotic, replicate populations evolved on CTT-HA diversified much more than the CTT-SA populations (Fig. 5a and Supp. Fig. 7a, b), a pattern corroborated by the average variances of individual-trait values across populations (Fig. 5b). Fruiting body counts and fruiting body area explained most of the inter-population diversification observed for CTT-HA in TS-abiotic (Fig. 5b and Supp. Fig. 1g). Evolution on CTT-SA resulted in extremely low levels of inter-population diversity at the end of MyxoEE-3, which was not significantly greater than diversity among the Anc founding clones (Fig. 5a and Supp. Fig. 7a, b), despite the extensive collective morphological divergence of the CTT-SA treatment away from its ancestral state (Fig. 2b, c and Supp. Fig. 1e, f). This lack of diversification is due to the systematic parallel loss of developmental proficiency by all six populations examined in this treatment (Fig. 2a, b, d and Supp. Fig. 1g).

Stochastic morphological diversification was also indirectly limited by specific TS-biotic selective environments. Evolution in the presence of either prey species on CTT hard agar allowed greater latent diversification of developmental phenotypes than evolution in the absence of prey, both across the collective morphospace (Fig. 5c and Supp. Fig. 7c, d) and often at single traits considered individually (Fig. 5d and Supp. Fig. 4g), as did adaptation to *E. coli* as prey on TPM agar. Stochastic diversification was lowest on *B. subtilis*-TPM, largely due to systematic parallel reductions of trait values across the four populations (Fig. 4a, c, d and Supp. Fig. 4g) (as occurred also in the CTT-SA populations of TS-abiotic (Fig. 2a, c, d and Supp. Fig. 1g).

Comparing treatments sharing the same prey species (*B. subtilis*-CTT vs *B. subtilis*-TPM; *E. coli*-CTT vs *E. coli*-TPM), *M. xanthus* diversification was lower in the TPM treatments than in the CTT treatments (Fig. 5c). Collectively, these results demonstrate that stochastic developmental diversification mediated by LPE can be strongly influenced by both abiotic and biotic components of the selective environment, here surface stiffness and prey environment, respectively.

## Discussion

Phenotype expression and form often depend greatly on ecological context. Genetic causation of phenotypic novelty can thus evolve latently (hidden from selection) in one environment until the causal genotype is exposed to a different, inductive environment that triggers phenotypic manifestation. Here we have shown how both abiotic and biotic ecological factors—swarming-surface stiffness and predator–prey interactions, respectively—shape and limit the evolution of latent developmental phenotypes in the multicellular bacterium *M. xanthus*.

The sophisticated genetic and behavioral complexity of fruiting body development among myxobacteria clearly points to these forms of aggregative multicellularity conferring significant evolutionary benefits. But what those benefits are remain unclear and may include protection from abiotic stresses and biotic dangers (e.g., potential predation by nematodes[100]), enhanced dispersal, and positive effects of high density during spore germination and post-germination growth and predation[41]. Even less clear is any

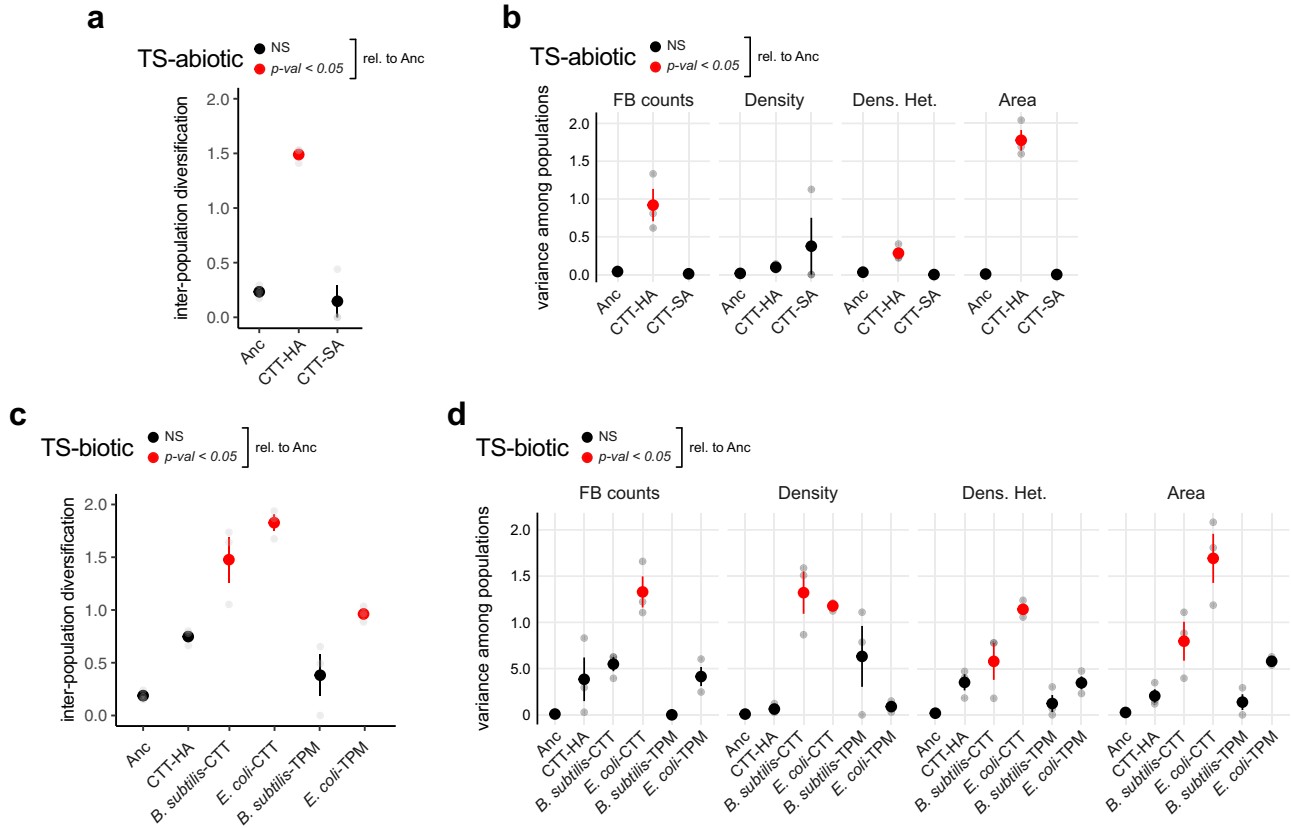

**Fig. 5 Deterministic limitation of stochastic latent-phenotype diversification by MyxoEE-3 selective environments. a, c** Morphological diversity among replicate TS-abiotic (**a**) and TS-biotic (**c**) populations evolved in the same selective environment compared to diversity among Anc subclones. **b, d** Variance of single morphological traits distributions calculated across TS-abiotic (**b**) and TS-biotic (**d**) evolved populations and across the Anc subclones. In all cases, large circles represent mean values ± SEM of three independent biological replicates (gray circles, $n = 3$), while red and black circles indicate significant ($p < 0.05$) and non-significant (NS) differences from Anc levels, respectively. (Significance was calculated in all cases with one-way ANOVA followed by two-tailed Tukey tests. $p$ values of all comparisons of evolved treatments with Anc, as well as all pairwise comparisons between evolved treatments, are reported in Data Fig. 5).

adaptive significance to the striking diversification of fruiting body morphology observed across myxobacterial species (or similarly across aggregative developmental species of dictyostelids)[41,42]. One evolution experiment—MyxoEE-7—has identified selective forces acting on fitness during *M. xanthus* development that can drive morphological diversification, namely selection mediated by distinct social partners such as cooperators, antagonists, or cheaters[11].

Our results with MyxoEE-3 suggest that evolutionary processes other than selection on fitness during development are also likely to play major roles in the morphological diversification of aggregative developmental systems. Ecological differences during growth without development—including a simple physical difference in growth surface and a difference in the identity of one-species prey environments - are found to evolutionarily shape not only details of fruiting body morphology in environments that do induce development but the very evolutionary persistence of development itself (Figs. 2a–c, 4a–c). Further investigations of how evolutionary forces other than selection on development per se interact with such selection to impact fruiting body evolution, including increases in morphological complexity, will be of interest.

The relative origins and long-term evolutionary integration of the genetic systems enabling the motility, predation, aggregative development, and germination components of myxobacterial multicellular life cycles remain to be thoroughly characterized[101,102]. However, the pervasiveness of LPE emergent from MyxoEE

evolution experiments[33,34,60,61,63,64,79,96,103,104] and of pleiotropy across *M. xanthus* behaviors from mutations engineered or induced in mechanistic molecular studies[62,105] together indicate that many loci contribute to more than one of these behaviors. Increasingly systematic and extensive investigations of pleiotropy across *M. xanthus* behaviors under standardized conditions will provide greater insight into the shared vs. modular components of the genetic systems underlying these behaviors and their evolution[59].

When selection operates across whole myxobacterial life cycles, trade-offs mediated by antagonistic pleiotropy that may exist between adaptive improvement at any one life-history stage (e.g., population growth fueled by predation) and another (e.g., development) must be balanced[58]. Whether such trade-offs even exist and the detailed manner in which selection balances those that do may often depend on details of the ecological context within which those life cycles are played out. Suggesting that this will often be the case in natural contexts, MyxoEE-3 reveals profound differences in the character of developmental LPE as a function of the ecological details of vegetative growth, for example, the presence vs absence of prey and the identity of prey.

That adaptation to distinct prey environments when selection on development is relaxed often has large divergent effects on development suggests that prey environments will also impact developmental-system divergence when selection across whole life cycles favors high proficiency at both predation and development. To address this question, evolution experiments might be performed that include both predatory growth and starvation-

induced development within each selection cycle and multiple treatments that differ only in prey environments (perhaps with respect to both prey identity and prey-community complexity). Distinct prey environments may drive divergence at nutritional triggers of development, developmental timing, overall patterns of developmental gene expression, sporulation level, mechanisms of cheater-cooperator coevolution (for developmental cheating as well as cheating on cooperative predation traits), the relationship between fruiting body formation and sporulation, fruiting body morphology, and the molecular triggers and social character of spore germination to reinitiate predatory growth, and might even ultimately drive diversification of the very gene sets necessary for development[84,102]. Other experiments might be performed to test how, reciprocally, selection on different developmental traits (e.g., developmental timing or spore quality) might differentially impact predatory performance across various prey environments.

In the first evolution experiment with *M. xanthus* (MyxoEE-1), most replicate populations adapting to growth in nutrient-rich CTT liquid latently evolved large decreases in developmental proficiency, including at both fruiting body formation and sporulation[96]. However, the relative contributions of nutrient abundance vs the absence of a solid growth substratum, if any, to promoting the evolutionary degradation of development in MyxoEE-1 have been unclear. Our analyses of MyxoEE-3 suggest that mere relaxation of selection for developmental proficiency by the provision of abundant growth resources is often less important to the evolutionary fate of development than other details of the ecological context in which population growth occurs. Development was largely retained when growth under nutrient abundance was accompanied by swarming that employs both *M. xanthus* motility systems (on hard agar), whereas it tended to be lost when S-motility alone (on soft agar) (Fig. 2) or no motility at all (see ref. [96]) was employed during growth (Supp. Table 2). Developmental phenotypes were retained at much higher levels when *E. coli* was consumed as prey than when *B. subtilis* was consumed, regardless of abiotic context (Fig. 4).

Comparing only the TS-abiotic treatments, it appears that a mere difference in the relative employment of the two *M. xanthus* motility systems during growth determines the evolutionary fate of development. Evolution while swarming almost exclusively by S-motility on soft agar (at least in the ancestral state) led to systematic severe losses of developmental proficiency. In contrast, swarming on a hard-agar surface while more equally employing both A-motility and S-motility[80] led to only sporadic and relatively mild decreases in developmental phenotypes (Fig. 2). Intriguingly, the simple difference in surface stiffness between the CTT-HA vs CTT-SA regimes also led to selection on different genes within the same regulatory pathway controlling multicellular development (*lonD*/*bsgA*). Future analysis of the precise effects of mutations in these genes (and other genes mutated in parallel during MyxoEE-3) on fitness during swarming and on developmental phenotypes would likely provide novel insights into pleiotropic connections between development and the distinct mechanisms of A- and S-motility employed during vegetative growth.

Considering the TS-abiotic results, the different LPE effects of distinct prey environments among the TS-biotic treatments might be due to the differential effects of those prey environments on the relative employment of the two *M. xanthus* motility systems. Alternatively, differences in predator–prey interactions unrelated to motility may be at play.

Our experiments reveal both deterministic and stochastic forms of latent-phenotype diversification (LPD). In deterministic LPD, populations diversified systematically at the treatment level due to differences in average pleiotropic effects on development caused by adaptive mutations that arose in different growth

environments. In stochastic LPD, replicate populations within the same treatment diverged from one another over time, an outcome explained by stochastic variation across replicate evolving populations in the identity and/or temporal order of mutations that occurred within each. While such stochastic LPD is evident already from examining individual traits in single environments (Supp. Figs. 1g, 4g), perhaps its most striking manifestation in our data set is the remarkable diversification of developmental reaction norms among the CTT-HA populations (Fig. 3b and Supp. Fig. 2). Among these populations, the very sign of reaction-norm slopes diversified stochastically for all four examined morphological traits across part or all of the examined environmental gradient (Fig. 3b). Phenotypic plasticity is well recognized as often being important for fitness, evolutionary trajectories, and evolvability across variable environmental conditions[106]. Latently evolved reaction-norm diversity might fuel novel routes of adaptive innovation during future evolution through changing environments[89,90].

The likely contributions of LPE to long-term evolutionary processes have been receiving increasing attention[33–37,40]. The outcomes of MyxoEE-3 reported here suggest that not only LPE per se, but the ecological context in which LPE originates, may often be important for the evolution of many developmental systems.

## Methods
**Semantics**. Here we specify our intended meanings of several terms in the context of this study:

- *development* (with respect to myxobacteria) - the collective set of behavioral processes leading to both fruiting body morphogenesis and sporulation.
- *divergence* - any degree of evolved genetic or phenotypic difference, whether between ancestral and derived genotypes or between distinct derived genotypes.
- *diversification* - the process by which distinct contemporary individuals or populations diverged from one another evolutionarily, genotypically, and/ or phenotypically.
- *latent phenotype*[33,34] - a phenotype that is potentiated by a genotype but remains unmanifested until the causal genotype is exposed to an inductive environment. We avoid the term "cryptic genetic variation" because it (i) focuses on within-population variation rather than evolutionary processes over time, which might result in the fixation of alleles potentiating latent phenotypes and (ii) is frequently associated with selectively neutral alleles[37,40,107] whereas latent phenotypes might be potentiated by alleles that increased due to selection. See ref. [34] for elaboration on use of the term "latent-phenotype evolution" (LPE). Similar terminology has also been used recently by other authors (see ref. [35]).
- *inductive environment* - an environment that induces manifestation of a phenotype that is unmanifested in other (non-inductive) environments.

**MyxoEE-3**. The evolved populations used in this study are a subset of populations from a broader evolution experiment named MyxoEE-3 (Fig. 1a), where 'MyxoEE' stands for "Myxobacteria Evolution Experiment" and "3" refers to the temporal rank position of the first publication from MyxoEE-3[78] relative to those from other MyxoEEs[33]. Details of MyxoEE-3 have been described previously[11,33,34,64,78,79] and aspects of this experiment most relevant to this study have been summarized in the Introduction (Fig. 1b). We also provide an additional summary of the manipulated differences between the selective environments of TS-biotic (Supp. Table 2). Replicate MyxoEE-3 populations examined here were founded by distinct subclones of *M. xanthus* strain GJV1[69] (here referred to as "Anc" for ancestor), including six replicate populations each for the CTT-HA and CTT-SA treatments at cycle 40 and four each for the four treatments with prey and the CTT-HA treatment at cycle 18 (Supp. Table 1).

**Induction of development**. Populations of exponentially growing cultures were pelleted and resuspended in TPM liquid buffer[108] to a final density of ~$5 \times 10^9$ cells/mL. Starvation plates were prepared one day prior to the experiment by pouring 10 ml of TPM agar into small Petri dishes and allowing solidification while uncovered under laminar flow. In most developmental assays, the agar concentration of TPM agar was 1.5%, but we also manipulated agar concentration to include 0.5% and 1.0% when characterizing developmental plasticity in TS-abiotic evolved populations (Fig. 2 and Supp. Fig. 4). Consistently in all assays, 50 µl of

resuspended culture (~2.5 × 10⁸ cells) were spotted at the center of TPM agar plates and incubated at 32 °C for 5 days before the plates were imaged and morphological traits subsequently quantified.

**Image acquisition and trait quantification**. Starvation plates of the evolved populations were imaged after 5 days of starvation for representative pictures (Figs. 1a, 3a and Supp. Fig. 3b) with a Zeiss STEMI 2000 microscope and captured with a Nikon Coolpix S10 camera. Images for quantitative morphological analysis were acquired using an Olympus SXZ16 microscope mounting an Olympus DP80 camera system. The image-acquisition parameters were kept identical in all cases (exposure time = 9.9 ms, lens = Olympus 0.5xPF, zoom = 1.25x, ISO = 200, illumination = BF built-in system). Images were processed and analyzed following the protocol described in ref. [11]. Fruiting body morphology was characterized by measuring the following traits. *Fruiting body number*: total number of fruiting bodies on a single developmental plate; *Density*: gray-value intensity of pixels per fruiting body; *Density heterogeneity*: standard deviation of within fruiting body pixel-gray values (fruiting body density); *Fruiting body area*: plate-surface area occupied per fruiting body, expressed in total pixel number. For more details about each trait, see ref. [11]. Median values of *Density*, *Density heterogeneity*, and *Area* per plate were used for further analysis. Image acquisition and trait quantification of the evolved populations were always run in parallel with Anc during each biological replicate.

**Spore counts**. A total of ~2.5 × 10⁸ cells in a 50 μl suspension (TPM liquid) were spotted on each starvation plate (TPM 1.5% agar) and harvested after 5 days using a sterile spatula and washed into one milliliter of ddH₂O. Samples were then heated at 50 °C for 2 h to kill vegetative cells, sonicated, diluted into CTT soft agar, and incubated for 7 days before colonies were counted. Also, in these assays, ancestral subclones (Anc) and evolved populations were assayed in parallel within each biological replicate.

**Swarming assay**. The swarming rates of all TS-biotic evolved populations and their ancestors were quantified by measuring the total area covered by the myxobacteria swarm plated onto the original MyxoEE-3 selective environment. The experimental conditions were equivalent to those used during MyxoEE-3, except that in this study, Bacto agar instead of Select agar was used (ref. [104]). Plates containing either CTT-HA or TPM-HA were prepared 24 h prior to inoculation with *M. xanthus*. The two prey *E. coli* and *B. subtilis* were grown in CTT liquid to stationary phase before being spread over the plates. In all cases, 10 μl of myxobacteria cultures growing in CTT liquid were spotted on the plate at a final density of 5 × 10⁹ cells/ml. Plates were incubated at 32 °C with 90% rH for 4 days, after which swarm areas were manually outlined and subsequently measured with the software ImageJ[109].

**Statistical analysis**. All experiments were performed in three temporally independent replicate blocks. Each replicate consisted of the analyzed evolved populations and their respective ancestral Anc subclones. R v4.0.0 software was used for all statistical analyses[110]. Spore counts were log₁₀-transformed prior to statistical analysis. For those cases in which the number of spores detected at the lowest dilution factor was zero (0), these counts were converted to one (1) prior to the log₁₀-transformation.

**Morphological analysis**. Using a previously developed approach[11], fruiting body morphology was analyzed with multivariate analysis. In brief, the obtained morphological-trait values were first averaged across all evolved population replicates from each selective environment and across the Anc-subclones for each biological assay replicate. Then, *stat::prcomp()* function in R was used to run PCA on the morphological scaled values. Plots reporting the PCA results (referred as morphospace in the main text) were obtained using the *ggbiplot::ggbiplot()* (Figs. 1b, 3b) or *ggplot2::ggplot()* (Supp. Fig. 3c) functions. Only in the case of CTT-SA PCA results, replicate three (3) was adjusted by a small false value of 0.05 exclusively for graphical purposes in Fig. 1b. This adjustment was not performed for any of the other analyses that focused on CTT-SA. After assessing the homoscedasticity of the data's dispersion in the multivariate space with the *vegan::betasiper()* function followed by a post hoc Tukey (*stats::TukeyHSD()* function), perMANOVA (*vegan::adonis()* function) was used to test whether selective environments and ancestral identity (Anc) significantly structured the data's dispersion. Inter-population diversification was calculated as in ref. [11].

Our measure of morphological integration was obtained from the covariance matrix of the four morphological traits and expressed as the variance of their eigenvalues $var(\lambda)$[82,83]. Covariance matrices of the morphological traits were produced for each treatment individually. Eigenvalue variances were obtained and scaled to the theoretical maximum using *evolqg::CalcEigenVar()* in R[83,111].

**Cluster analysis**. Cluster numbers in k-means analyses were determined by estimating the average silhouette width from the PCA results (*factoextra::fviz_nbclust()*). Once the number of optimal clusters in each case was defined (k = 2 in both TS-abiotic and TS-biotic), the actual k-means analysis was performed (*stats::kmeans()*)

and results plotted (*factoextra::fviz_clust()*) (Supp. Figs. 1d–f, 4d–f). In addition, hierarchical-clustering analysis based on the *ward* method was also calculated from the relative Euclidean distances between all treatments' centroids mapped on the morphospace (*stats::hclust()*) (Supp. Figs. 1f, 4f).

**Reporting summary**. Further information on research design is available in the Nature Research Reporting Summary linked to this article.

## Data availability

All datasets are available at figShare (figshare.com/s/fbf02ac33fc8c93b59f7). Raw images and programming codes used for the analyses are available upon request to the corresponding author.

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

## Acknowledgements
We thank Kaitlin Schaal for her comments on the manuscript. This work was funded in part by two EMBO Long-Term Fellowships, ALTF 1208-2017 to M.L.F. and ALTF 1411-2012 to O.R. and Swiss National Science Foundation (SNF) grant 310030B_182830 to G.J.V.

## Author contributions
M.L.F., O.R., and G.J.V. designed the research and revised the manuscript; M.L.F. and G.J.V. drafted the manuscript; M.L.F., O.R., and H.K. conducted the experiments; M.L.F. analyzed the data and created the figures.

## Competing interests
The authors declare no competing interests.
