## [Peer Review File · Communications Biology]

Reviewers' comments:

Reviewer #1 (Remarks to the Author):

I have read the article Fortezza et al. , examining the potential for the evolution of latent traits in response to an interesting lab selection experiment. The authors have performed a series of experiments in *M. xanthus* that can induce evolutionary and plastic changes in four traits. The authors perform statistical analysis on these traits using a PCA in combination with a permanova, as well as a cluster analysis.

While I'm familiar with the motivation for such a study, and how it may be able to address its core ideas through stats, I am not experienced with lab experiments on bacteria. So please forgive my lack of clarity if it comes up in places. Overall, I think this manuscript has promise but in places it is difficult to follow the motivation for the different experiments. It may be that it would benefit from broader contextualization.

The idea that latent traits can evolve (and be important variation for adaptation) is an interesting topic for investigation. However, I felt that some further ideas need to be considered to set up the statistical analysis. For example. Pleiotropy is mentioned and used as an explanation for some of the patterns observed but I don't feel convinced that it has been tested against a null. For example, pleiotropy could be indicated by statistics that indicate how relationships change or stay the same across experiments. This needs more careful consideration, as there is also the chance that reaction norms can evolve in coordination. I have provided more detailed comments below.

There also seems to be some misunderstanding about what the stats indicate. The permanova is a useful technique but it needs to be explained much more clearly how it actually tests for latent evolution. Start your paragraphs with biological statements about what is being tested and why this statistic fits the situation. The same goes for the k-means clustering, why would clustering address your hypotheses? Remind the reader of your hypotheses for your tests.

If there is indirect evolution of latent traits would it not be important to test for relationships between reaction norms? You have data that is tractable for evolution, and tractable for plasticity, but there seems to be no one to one test of how these relate. This is where statistical expectations that line up with a biological hypothesis would be key. As it is now I don't see a clear interpretation of the evolution of latent traits.

Please find more precise comments below:

Line 39- could also cite Van Valen, L. (1973). Festschrift. Science 180, 488.

Line 40 could cite Skulason et al. 2019, Biological Reviews

Line 58- I appreciate the semantics section. I'm wondering here if cryptic genetic variation should be mentioned, if this is actually what is described here, or are the authors thinking more broadly about how developmental systems contain interactions that evolve, perhaps in response to a current environment which in turn affects the response to an environmental change and subsequent phenotypes that are induced in a future environment. There is also some work in ants with latent castes that can be induced that would serve as a good example here (Rajakumar et al 2012). This at least shows the presence of latent phenotypes as a first principle, but latent phenotype evolution itself might be something that requires more direct examination. Perhaps the authors could clarify? There is also a recent special edition of Evolution and Development focused on developmental bias that contains some relevant papers for the ideas being discussed.

Line 62- I wonder if the authors have considered the term 'genetic causation' when in fact it seems that environment is determining whether an allele is neutral, adaptive, or deleterious, and when a phenotype is determined by an environment interacting with genetic variation within the context of a developmental system (and its inherent interactions). I appreciate this is full of semantics but wanted

to raise this with no expectation this is taken on.

Lines 76-8- it would be useful to clarify whether this is a mutant line where the causative mutations are identified, just thinking of a general audience (of which I am a member)

lines 108-112- I would think that the selection on non-morphological factors is actually a direct form of selection as well given the example. A phenotype that affects fitness it seems through morphology.

Line 118- it wasn't clear to me what latent phenotypes this system has, latent from what perspective? Are they not just a plastic species? What makes this different?

Line 129 – so in other words this study is looking at adaptive plasticity, and is determining whether selection on adaptive plasticity affects plasticity in response to a different environment. It could be thought of in this way as well it would seem, pleiotropy in the evolution of plastic responses.

Lines 30-50 it seems like this would belong in a methods section, rather it would be good to see a set of main questions and hypotheses referred to that are tested with these approaches. I think this section is confusing without broader conceptual context.

Line 166- the PCA would not perform a test, looking at the methods mentioned on line 170 the permanova would actually be a test that would reveal differences, this might be reflected in a scatterplot. There is a nuance here that needs to be paid attention to.

Line 170- PC1 would always explain the most variation, but it would be helpful to say how much % wise this is here. I see that PC1 explains over 84% of the variation, that would make it impossible for the other eigenvectors to contribute a similar amount of variation? (i.e. less than 16% of the variation is left after PC1).

Related to this the high explanatory power of PC1 suggests the four traits are highly integrated/correlated. There are a number of studies which use eigenvalue variance to investigate integration. What that means is that 1) pleiotropy is strong, you may want to consider whether eigenvalue variance changes among lines, it may say something about the effect of pleiotropy (i.e. if there is pleiotropy we may expect that it is stable between lines, if these traits are not pleiotropic you may see they become more independent in terms of how they change, a given line may show a decrease in eigenvalue variance, Pavlicev et al. 2009 evol biol)

line 178- it is valid to consider traits individually but it would help to explain the biological reasons for looking at them in two different ways.

Page 8 line 30, it would be helpful to define when it can be said that a trait evolved latently. Can a statistical test which identifies latent evolution be described. The figure seems to provide trends but I don't think there is a null hypothesis to go against. It also comes to mind here, given the focus on pleiotropy, whether the authors have considered looking more explicitly at correlations between reaction norms and how they have changed. There is a small literature on 'plasticity integration' which may be useful (recent paper by Navon et al. 2021, which cites other studies on plasticity integration that have been undertaken).

Reviewer #2 (Remarks to the Author):

This manuscript deals with latent evolution of fruiting body (FB) development in the social bacterium *Myxococcus xanthus*. Using bacterial strains that were previously experimentally evolved in different ecological conditions (that did not induce FB formation), the authors test if adaptation to some of these ecological parameters affects the evolution of FB morphology. They found that adaptation to substrate stiffness and prey identity did affect FB morphology and developmental plasticity, but also impacts on the variability of evolutionary trajectories among replicate populations. These results show that adaptation to a given environment can change phenotypic traits even if they are not expressed in this environment, and that shared genetic elements are involved in FB development, motility and predation. This work brings original, new insights on latent phenotypic evolution in bacteria, and I believe it will be of interest to researchers working in the field of experimental evolution, as well as to

many evolutionary biologists and microbiologists. Overall, the manuscript is convincing: experiments are well designed and analyzed, the level of details in the 'Methods' section is appropriate, and the conclusions are supported by the data. The manuscript is very well written, clear and easy to follow.

(1). My only main comment relates to the interpretation of the findings. In the introduction, the authors present a brief description of the MyxoEE3 experiment, but they do not mention the main phenotypic outcome(s) of this experiment. To which extend did the different lineages adapt to their respective selective environment? Was the magnitude of phenotypic evolution comparable in the different environments/lineages? Having some general information related to these questions will be important as it can change the way to think about the results. For example, could the loss of FB formation observed in CTT-SA treatment be due to faster adaptation under this treatment than in the CTT-HA treatment (possibly linked to stronger selective pressures in CTT-SA plates)? Would we also observe a loss of FB formation if lineages were passaged for additional cycles in CTT-HA? In other words, I wonder if the difference observed in the different treatments (including substrate stiffness and prey identity) are due to qualitatively divergent evolutionary trajectories in the different environments, or if they are just the result of a different speed in their respective adaptive walks. An analysis of possible correlations between fitness/motility gains in the selected environment (given that at least part of these data seems to be available from ref. 74) vs. developmental phenotypes would be most relevant in that respect.

Minor comments:

- (2) L191: Is the full list of mutations found in the sequenced clones available somewhere? I checked in ref. 74, but I could only find the list of mutations in genes related to motility in the supplementary material. It would be useful to have access to the complete list of mutations so that the interest reader may dig deeper in the possible molecular causes of phenotypic evolution. In particular, are there any interesting insights coming from the analysis of mutations in genes involved in motility? Although the authors rightfully focus on convergent mutations at the level of individual genes, it is also common to see convergent mutations at the level of signaling pathways. It would be interesting to know if the exact identity of mutations affecting motility in the different treatments may correlate with (or explain) some of the developmental phenotypes.
- (3) L193: "(Fig. S2, Table S3, see Methods)": I didn't find any mention of this analysis in the Methods section.
- (4) L272: "Fig. 1C": shouldn't the authors refer to Fig. 3C here?
- (5) L281: "two... clusters distinguished solely by the presence or absence of *B. subtilis*": the authors may want to reference Fig. S4E too at some stage in this paragraph.
- (6) L291-292: The statement "In *Myxococcus* literature, sporulation is frequently, if not always, associated positively with FB formation" should be referenced (but maybe references [37,86] found in the next sentence are misplaced?)
- (7) L440: "Myxo-EE3I": what does "I" stand for? Is that a typo?
- (8) Fig. S6: I am not expert of PCA analyses, but I wonder about the correspondence between data shown in panels S6C and S6D. In particular, the large variability between the *B. subtilis*-CTT populations shown in panel S6D (and also in Fig. 4C) does not really stand out when looking at panel S6C. And the inverse is true for *E. coli*-TPM populations. This might just be some visual effect due to the different types of representation used in the two panels, but could the authors double-check that there hasn't been any inadvertent inversion in the raw data used to generate these plots?

Reviewer #3 (Remarks to the Author):

1. This study follows up on an evolution experiment previously published by the authors, by carrying out a series of measurements of evolved populations of *M. xanthus*. This already published experiment had very clever design, but has quite a few treatments, and I think a simple figure explaining the experiment will go a long way to helping the reader understand the study- perhaps the authors have a

figure they could adapt from one of these previous papers? Basically, the authors measure a set of *M. xanthus* clones that evolved in two broad sets of conditions. The first set is called TS – A. In these treatments populations were evolved on two different types of agar that select for the maintenance of either one type of motility (S motility) or both S&A motility. The second set of treatments, TS-B, have one treatment that I couldn't discern from the description, and four other treatments where the nutrients for the evolving populations of *M. xanthus* was either *E. coli* and growth media or *B. subtilis* and growth media or just *E. coli* bacteria as food or just *B. subtilis* bacteria as food. The authors take strains evolved under these conditions for forty cycles and carry out five measurements. Four of them are of different aspects of the fruiting body and one of the measurements is of the spore.

2.The introduction explains how common eco evo devo is without providing an actual example. The authors have provided citations, and the introduction is very long, but please describe in clear language an example of a latent phenotype contributing to evolution, lets say one in *M. xanthus*, and one from another experimental system. Also, the introduction could do with some editing and economy- it tends to be a little repetitive- for example the authors repeat twice how powerful the *M. xanthus* system is.

3.The results are very hard to follow because this study uses so many acronyms. I know that they are defined them, but I keep having to go flip down the page to find 6 acronyms on in the first half page of the results that are specific to your experimental system- LPE, FB, TSA, TSB, CTA-HA, CTT-SA and some readers might need to look up PCA. You could write in some more descriptions to help the reader keep up with what is going on in the results.

4. Line 186. the authors assert that under high nutrient conditions adaptation to swarming on a soft surface traded off against general developmental proficiency where does this conclusion come from- it seems like a complete non-sequitor after what's been discussed- for example, what does developmental proficiency mean in this context?

5.Line 189 the authors introduce some sequence data described in reference 74, it would be good to mention that the populations were sequenced or clones were sequenced or whatever in the description of the experiment. The authors also introduced some new nomenclature here P29P31P33 which I guess refer to clones but they haven't actually been properly introduced.

6.Line 206 the authors provide speculation here about lonD having contributed towards developmental loss. What is developmental loss (its used once in the document without definition)? how did did the authors reach this conclusion?

7.Line 209 the authors conclude that because the genes hsfB and lonD are both in the same pathway they have a related function. While this might be true, the mutations in these two genes could be leading to completely different outcomes. This is probably the case since one treatment strongly selected for mutations in hsfB and the other selected for mutations in lonD. Are the mutations in these genes loss of function mutations, ie do they cause frameshifts or early stops? Also based on the understanding of the pathway is hsfB basically a negative regulator of lonD or positive regulator?

8.Line 222 the same sentence is repeated here with each repetition having a different citation.

9.Line 224 I found it very difficult to extract any meaning from this sentence. There's so much going on in here, you need to unpack it into a few different sentences so that you can clearly express these ideas, and provide the motivation of what you're going to do next.

10.Line 233 onward. It is first pointed out that for all four traits in populations varied in the slope and the sign of the slope, but is then mentioned (in parentheses) that the only plausible explanation for this is the number of mutations in the different clones. Then this phenomenon is referred to as "stochastic radiation". First, the different possible explanations for the observed patterns of reaction

norm variation need to be properly explicated, then you need to clearly explain why the accumulation of mutations is the best explanation. The writer is making too many leaps here and expecting the reader to keep up- also it's very difficult to assess any of the claims made here.

All major text changes in the manuscript are highlighted in blue.

Reviewers' comments:

Reviewer #1 (Remarks to the Author):

*I have read the article Fortezza et al. , examining the potential for the evolution of latent traits in response to an interesting lab selection experiment. The authors have performed a series of experiments in *M. xanthus* that can induce evolutionary and plastic changes in four traits. The authors perform statistical analysis on these traits using a PCA in combination with a permanova, as well as a cluster analysis.*

While I'm familiar with the motivation for such a study, and how it may be able to address its core ideas through stats, I am not experienced with lab experiments on bacteria. So please forgive my lack of clarity if it comes up in places. Overall, I think this manuscript has promise but in places it is difficult to follow the motivation for the different experiments. It may be that it would benefit from broader contextualization.

The idea that latent traits can evolve (and be important variation for adaptation) is an interesting topic for investigation. However, I felt that some further ideas need to be considered to set up the statistical analysis. For example. Pleiotropy is mentioned and used as an explanation for some of the patterns observed but I don't feel convinced that it has been tested against a null. For example, pleiotropy could be indicated by statistics that indicate how relationships change or stay the same across experiments. This needs more careful consideration, as there is also the chance that reaction norms can evolve in coordination. I have provided more detailed comments below.

There also seems to be some misunderstanding about what the stats indicate. The permanova is a useful technique but it needs to be explained much more clearly how it actually tests for latent evolution. Start your paragraphs with biological statements about what is being tested and why this statistic fits the situation. The same goes for the k-means clustering, why would clustering address your hypotheses? Remind the reader of your hypotheses for your tests.

If there is indirect evolution of latent traits would it not be important to test for relationships between reaction norms? You have data that is tractable for evolution, and tractable for plasticity, but there seems to be no one to one test of how these relate. This is where statistical expectations that line up with a biological hypothesis would be key. As it is now I don't see a clear interpretation of the evolution of latent traits.

We thank the reviewer for the many thoughtful comments. We respond to each of the more precise comments below.

Please find more precise comments below:

Line 39- could also cite Van Valen, L. (1973). Festschrift. Science 180, 488.

Line 40 could cite Skulason et al. 2019, Biological Reviews

The two references have been added.

Line 58- I appreciate the semantics section. I'm wondering here if cryptic genetic variation should be mentioned, if this is actually what is described here, or are the authors thinking more broadly about how developmental systems contain interactions that evolve, perhaps in response to a current

environment which in turn affects the response to an environmental change and subsequent phenotypes that are induced in a future environment. There is also some work in ants with latent castes that can be induced that would serve as a good example here (Rajakumar et al 2012). This at least shows the presence of latent phenotypes as a first principle, but latent phenotype evolution itself might be something that requires more direct examination. Perhaps the authors could clarify? There is also a recent special edition of *Evolution and Development* focused on developmental bias that contains some relevant papers for the ideas being discussed.

We agree and in the *Semantics* section under 'latent phenotypes' have included a clarification of why we do not refer to CGV in our study, namely for reasons already specified in previously published work. Our focus is not on CGV present in a given population at a given time, but rather we emphasize the processes over time by which phenotypes can evolve latently.

Line 62- I wonder if the authors have considered the term 'genetic causation' when in fact it seems that environment is determining whether an allele is neutral, adaptive, or deleterious, and when a phenotype is determined by an environment interacting with genetic variation within the context of a developmental system (and its inherent interactions). I appreciate this is full of semantics but wanted to raise this with no expectation this is taken on.

To avoid potential confusion we have changed the term from 'genetic causation of' to the 'genetic basis of' and modified the remainder of the sentence. **L51**

Lines 76-8- it would be useful to clarify whether this is a mutant line where the causative mutations are identified, just thinking of a general audience (of which I am a member)

It seems that the term 'modes' may have been interpreted differently than intended. Thus, we now use 'motility system' and mention that each system involves many different genes. We have included a new reference to support this statement. **L77-80**

lines 108-112- I would think that the selection on non-morphological factors is actually a direct form of selection as well given the example. A phenotype that affects fitness it seems through morphology.

We remove the term *direct* before *selection* to avoid potential confusion. **L111**

Line 118- it wasn't clear to me what latent phenotypes this system has, latent from what perspective? Are they not just a plastic species? What makes this different?

We use the term latent to refer to those traits that evolved invisibly while selection was acting on other traits. Hence, by definition, traits that evolved latently are those that remained *unexpressed* until they were induced by a new environment. In that specific sentence we tried to highlight the versatility of using *M. xanthus*, since the studied developmental process (fruiting body formation) is inducible and thus also easy to maintain unexpressed and hidden from direct selection. Instead of expanding the text even more, we have slightly modified the phrasing and have referred to the *Semantics* section.

Line 129 – so in other words this study is looking at adaptive plasticity, and is determining whether selection on adaptive plasticity affects plasticity in response to a different environment. It could be thought of in this way as well it would seem, pleiotropy in the evolution of plastic responses.

We would not describe the study this way. Plasticity is only a minor theme of the paper. We later discuss plasticity only in the context of how developmental morphology of a given genotype or population shifts as a function of agar concentration under otherwise identical starvation conditions. MyxoEE-3 was not designed to select for adaptive plasticity at any phenotype. Rather, we test whether adaptation to any of the MyxoEE-3 selective environments examined in this study indirectly results in evolutionary change in developmental phenotypic plasticity with respect to how agar concentration impacts developmental morphology. Because evolving populations did not undergo development, the

evolutionary change in developmental plasticity across variable agar concentrations was clearly *non-adaptive*.

Lines 30-50 it seems like this would belong in a methods section, rather it would be good to see a set of main questions and hypotheses referred to that are tested with these approaches. I think this section is confusing without broader conceptual context.

While we understand the suggestion to move this part, we consider this design and methodology overview to be very important for introducing the study as a whole. However, considering this comment, we restate the general hypothesis at the beginning of the paragraph and have sought to make the phrasing of the paragraph more accessible. **L134-L135**

Line 166- the PCA would not perform a test, looking at the methods mentioned on line 170 the permanova would actually be a test that would reveal differences, this might be reflected in a scatterplot. There is a nuance here that needs to be paid attention to.

We have rephrased to emphasize the role of the perMANOVA test in inferring treatment-level differentiation **L177-L179, L281-L285**

Line 170- PC1 would always explain the most variation, but it would be helpful to say how much % wise this is here.

We agree, and we add this information also in the main text in addition to the main figure(s). **L180, L287**

I see that PC1 explains over 84% of the variation, that would make it impossible for the other eigenvectors to contribute a similar amount of variation? (i.e. less than 16% of the variation is left after PC1).

We recognize the mistake in the language used and have corrected this accordingly **L180-181**

Related to this the high explanatory power of PC1 suggests the four traits are highly integrated/correlated. There are a number of studies which use eigenvalue variance to investigate integration. What that means is that 1) pleiotropy is strong, you may want to consider whether eigenvalue variance changes among lines, it may say something about the effect of pleiotropy (i.e. if there is pleiotropy we may expect that it is stable between lines, if these traits are not pleiotropic you may see they become more independent in terms of how they change, a given line may show a decrease in eigenvalue variance, Pavlicev et al. 2009 evol biol)

We thank the reviewer for suggesting this type of analysis, which we find very interesting. We have adopted the suggested approach (based on Pavlicev et al. 2009 and Machado et al. 2019) to test whether the degree of integration of the focal traits evolved differently in the different MyxoEE-3 treatments. We discuss the obtained results in the main text (**L183-L189, L289-L298**), add two panels to the two supplementary figures reporting the multivariate analyses (**Fig S2C**, and **Fig S4C**) and describe the methodology in the Methods section under *Morphological analysis*. The most interesting results emerge from the TS-biotic data, which strongly suggest that different forms of selection imposed by distinct MyxoEE-3 ecological contexts - including prey identity – can differentially impact trait correlations, and thus likely the degree of pleiotropic connectedness among traits (Wagner and Zhang 2011).

line 178- it is valid to consider traits individually but it would help to explain the biological reasons for looking at them in two different ways.

We now explain why we look at single traits. **L196 – L197**

Page 8 line 30, it would be helpful to define when it can be said that a trait evolved latently. Can a statistical test which identifies latent evolution be described. The figure seems to provide trends but I

don't think there is a null hypothesis to go against. It also comes to mind here, given the focus on pleiotropy, whether the authors have considered looking more explicitly at correlations between reaction norms and how they have changed. There is a small literature on 'plasticity integration' which may be useful (recent paper by Navon et al. 2021, which cites other studies on plasticity integration that have been undertaken).

That a trait evolved latently can be strongly claimed only when it is known that the genetic basis of a focal evolved phenotype evolved prior to the environment-specific expression of that phenotype. Thus, detailed knowledge of evolutionary histories is required, knowledge allowed by designed evolution experiments such as MyxoEE-3. If the relevant environmental histories of evolutionary lineages are sufficiently known, then the statistical aspect is straightforward. It then simply involves testing whether a trait that was not expressed during evolution over a known period in a known environment has significantly different values between ancestral vs evolved genotypes when expression of that trait is induced in a relevant environment.

We agree that the suggested analysis on plasticity integration is interesting, however, we believe it is beyond the scope of our manuscript. For the main point we want to make on plasticity, we don't think it necessary to explore additional null-hypotheses other than the one already tested for the obtained reaction norms (i.e., H_0 : reaction norm Anc = reaction norm Evolved). The rejection of H_0 demonstrates latent evolutionary change in reaction norms because development was not under selection during the evolution experiment. This is the same reasoning used to demonstrate the latent evolution of single morphological traits examined in one inductive environment, except here we're considering the reaction norm of a phenotype across variable environments as the focus of analysis.

Reviewer #2 (Remarks to the Author):

*This manuscript deals with latent evolution of fruiting body (FB) development in the social bacterium *Myxococcus xanthus*. Using bacterial strains that were previously experimentally evolved in different ecological conditions (that did not induce FB formation), the authors test if adaptation to some of these ecological parameters affects the evolution of FB morphology. They found that adaptation to substrate stiffness and prey identity did affect FB morphology and developmental plasticity, but also impacts on the variability of evolutionary trajectories among replicate populations. These results show that adaptation to a given environment can change phenotypic traits even if they are not expressed in this environment, and that shared genetic elements are involved in FB development, motility and predation. This work brings original, new insights on latent phenotypic evolution in bacteria, and I believe it will be of interest to researchers working in the field of experimental evolution, as well as to many evolutionary biologists and microbiologists. Overall, the manuscript is convincing: experiments are well designed and analyzed, the level of details in the 'Methods' section is appropriate, and the conclusions are supported by the data. The manuscript is very well written, clear and easy to follow.*

(1). My only main comment relates to the interpretation of the findings. In the introduction, the authors present a brief description of the MyxoEE3 experiment, but they do not mention the main phenotypic outcome(s) of this experiment. To which extent did the different lineages adapt to their respective selective environment? Was the magnitude of phenotypic evolution comparable in the different environments/lineages? Having some general information related to these questions will be important as it can change the way to think about the results. For example, could the loss of FB formation observed in CTT-SA treatment be due to faster adaptation under this treatment than in the CTT-HA treatment (possibly linked to stronger selective pressures in CTT-SA plates)? Would we also observe a loss of FB formation if lineages were passaged for additional cycles in CTT-HA? In other words, I wonder if the difference observed in the different treatments (including substrate stiffness and prey identity) are due to qualitatively divergent evolutionary trajectories in the different environments, or if they are just the result of a different speed in their respective adaptive walks. An analysis of possible

correlations between fitness/motility gains in the selected environment (given that at least part of these data seems to be available from ref. 74) vs. developmental phenotypes would be most relevant in that respect.

The fundamental question raised by the reviewer regards *how* differences in MyxoEE-3 environments cause differences in the degree of latent evolution of developmental traits. This question is very interesting and we are pleased to address it with additional analyses and results.

We now explicate two hypotheses regarding how the different MyxoEE-3 treatments may have caused large treatment-level differences in the degree of developmental-phenotype evolution.

- One hypothesis is that the rate of genomic evolution may have varied across treatments in a manner and to a degree that explains differences in the rate of developmental-phenotype evolution. Differences in the rate of genomic evolution might in turn be due to differences in selection strength (as the reviewer notes) and/or to differences in mutation supply caused by differences in the average degree of per-cycle population growth across treatments.
- The second hypothesis is that distinct sets of mutations selectively favored by the various MyxoEE-3 treatments differ, on average, in their effects on developmental phenotypes.

To evaluate these hypotheses we use two approaches, both of which are included in the revised manuscript. For the TS-abiotic treatments we use published mutation data to show that the two treatments did not differ significantly in the number of mutations found in evolved clones despite differing dramatically in their average developmental phenotypes (Figs. 2, 3). Clones sequenced from the CTT-SA populations had on average 14.4 mutations (excluding one mutator clone) and those from the CTT-HA treatment had 11.7 on average, but this difference was not significant. Importantly though, even if the difference were significant, such a small difference of ~25% in the rate of genomic evolution would not come close to explaining the much larger differences observed in the degree of developmental-phenotype evolution between the CTT-HA and CTT-SA treatments (Fig. 2).

For the TS-biotic treatments, we now include swarming-rate data and use relative swarm sizes as a proxy for relative population sizes achieved in each MyxoEE-3 environment to address whether variable rates of genomic evolution might explain the observed outcomes. These new results demonstrate that the degree of phenotypic change at the focal developmental traits does not correlate with either i) estimates of relative average population size achieved within MyxoEE-3 cycle as represented by swarm sizes in the respective environments or ii) the degree of evolutionary change in swarming rates in the selective environment of each MyxoEE-3 treatment. Our results therefore strongly support the second hypothesis above. We now discuss these points in the main text (**L345-L384**) and include these new results in a supplementary figure (**Fig S6**).

Minor comments:

- (2) L191: *Is the full list of mutations found in the sequenced clones available somewhere? I checked in ref. 74, but I could only find the list of mutations in genes related to motility in the supplementary material. It would be useful to have access to the complete list of mutations so that the interest reader may dig deeper in the possible molecular causes of phenotypic evolution. In particular, are there any interesting insights coming from the analysis of mutations in genes involved in motility? Although the authors rightfully focus on convergent mutations at the level of individual genes, it is also common to see convergent mutations at the level of signaling pathways. It would be interesting to know if the exact identity of mutations affecting motility in the different treatments may correlate with (or explain) some of the developmental phenotypes.*

We add the list of all mapped mutation in a new supplementary table (Table S4)

- (3) L193: "(Fig. S2, Table S3, see Methods)": I didn't find any mention of this analysis in the Methods section.

Thanks for catching this. We removed the reference 'see Methods' since it was misplaced.

- (4) L272: "Fig. 1C": shouldn't the authors refer to Fig. 3C here?

Indeed, the correct reference is to Fig 3B, 3C (now Fig 4B, 4C). We changed this accordingly. **L309**

- (5) L281: "two... clusters distinguished solely by the presence or absence of *B. subtilis*": the authors may want to reference Fig. S4E too at some stage in this paragraph.

True, the reference to Fig S4E has been added. **L318**

- (6) L291-292: The statement "In *Myxococcus* literature, sporulation is frequently, if not always, associated positively with FB formation" should be referenced (but maybe references [37,86] found in the next sentence are misplaced?)

Correct, the two references 37 and 86 are misplaced and are linked to the mentioned statement on sporulation. We changed the text accordingly. **L330**

- (7) L440: "Myxo-EE31": what does "I" stand for? Is that a typo?

Yes, "I" is a typo and it has been removed. **L520**

- (8) Fig. S6: I am not expert of PCA analyses, but I wonder about the correspondence between data shown in panels S6C and S6D. In particular, the large variability between the *B. subtilis*-CTT populations shown in panel S6D (and also in Fig. 4C) does not really stand out when looking at panel S6C. And the inverse is true for *E. coli*-TPM populations. This might just be some visual effect due to the different types of representation used in the two panels, but could the authors double-check that there hasn't been any inadvertent inversion in the raw data used to generate these plots?

We double-checked the data entries and re-performed the entire analysis and obtained the same results. We think that such apparent incongruity is caused by two factors. 1 - All plots in **Fig S7D** (previously Fig S6D) are based on distance matrices calculated on all four dimensions from the PCA. Thus, a direct comparison is a bit difficult to make on the plots in **Fig S7C** (previously Fig S6C) because only two of the four dimensions are reported. Moreover, the two axes in **Fig S7C** reporting PC1 and PC2 values have different ranges. Thus, differences on PC2 (from -1 to 1) are in fact less prominent than differences on PC1 (from -2 to 3) and thus differences between line lengths could be deceiving from the reported measurements in **Fig S7D**. However, we have remade the graphs using the same ranges for both axes in **Fig S7A and S7C** to help the reader to go through both C and D panels more easily.

Reviewer #3 (Remarks to the Author):

1. This study follows up on an evolution experiment previously published by the authors, by carrying out a series of measurements of evolved populations of *M. xanthus*. This already published experiment had very clever design, but has quite a few treatments, and I think a simple figure explaining the experiment will go a long way to helping the reader understand the study- perhaps the authors have a figure they could adapt from one of these previous papers? Basically, the authors measure a set of *M. xanthus* clones that evolved in two broad sets of conditions. The first set is called TS – A. In these treatments populations were evolved on two different types of agar that select for the maintenance of either one type of motility (*S* motility) or both *S*&*A* motility. The second set of treatments, TS-B, have one treatment that I couldn't discern from the description, and four other treatments where the nutrients for the evolving populations of *M. xanthus* was either *E. coli* and growth media or *B. subtilis* and growth media or just *E. coli* bacteria as food or just *B. subtilis* bacteria as food. The authors take strains evolved under these conditions for forty cycles and carry out five

measurements. Four of them are of different aspects of the fruiting body and one of the measurements is of the spore.

We thank the reviewer for the suggestion. We introduce a new Figure 1 that illustrates MyxoEE-3 (Fig 1A) and the specific selective environments that were analyzed in our study (Fig. 1B). Moreover, we also add a panel 1C to explicitly refer to the clear distinction between the selective and inductive environments defined by our experimental approach.

2. The introduction explains how common *eco evo devo* is without providing an actual example. The authors have provided citations, and the introduction is very long, but please describe in clear language an example of a latent phenotype contributing to evolution, let's say one in *M. xanthus*, and one from another experimental system.

We introduce two examples about the induction of two classical examples of latent phenotypes that have been previously described (Lédon-Retting et al 2010, Berger et al 2011) L54-L57. However, for *M. xanthus* we left the same references that we have reported originally.

Also, the introduction could do with some editing and economy- it tends to be a little repetitive- for example the authors repeat twice how powerful the *M. xanthus* system is.

We have trimmed the section addressing the *M. xanthus* system while retaining the most relevant information.

3. The results are very hard to follow because this study uses so many acronyms. I know that they are defined them, but I keep having to go flip down the page to find 6 acronyms on in the first half page of the results that are specific to your experimental system- LPE, FB, TSA, TSB, CTA-HA, CTT-SA and some readers might need to look up PCA. You could write in some more descriptions to help the reader keep up with what is going on in the results.

We agree with this comment, although we found it difficult to remove all acronyms, especially in the case of treatment names. Also, we retain the short term LPE since it has been used in multiple already published (Rendueles et al 2020, Freund et al 2021). However, we have changed:

FB to fruiting body

TS-A to TS-abiotic

TS-B to TS-biotic

4. Line 186. the authors assert that under high nutrient conditions adaptation to swarming on a soft surface traded off against general developmental proficiency where does this conclusion come from- it seems like a complete non-sequitor after what's been discussed- for example, what does developmental proficiency mean in this context?

We have made that sentence more specific to make our conclusion clearer. L235-L237

5. Line 189 the authors introduce some sequence data described in reference 74, it would be good to mention that the populations were sequenced or clones were sequenced or whatever in the description of the experiment. The authors also introduced some new nomenclature here P29P31P33 which I guess refer to clones but they haven't actually been properly introduced.

In lines 209-211 we did specify the following:

"we examined previously reported mutation profiles of clones isolated from each of the CTT-HA and CTT-SA cycle-40 populations for possible candidate mutations".

Given the comment on the definition of 'P', we introduce in L218 a clearer reference to its meaning. Similarly, we introduce the same specification in the caption of Table S3.

6. Line 206 the authors provide speculation here about *lonD* having contributed towards developmental loss. What is developmental loss (its used once in the document without definition)? how did the authors reach this conclusion?

We changed the text, and we now refer more precisely to “the loss of fruiting body development”
L226

7.Line 209 the authors conclude that because the genes hsfB and lonD are both in the same pathway they have a related function. While this might be true, the mutations in these two genes could be leading to completely different outcomes. This is probably the case since one treatment strongly selected for mutations in hsfB and the other selected for mutations in lonD. Are the mutations in these genes loss of function mutations, ie do they cause frameshifts or early stops? Also based on the understanding of the pathway is hsfB basically a negative regulator of lonD or positive regulator?

We did refer to the known molecular interaction between HsfB and LonD through HsfA. However, to address the specific nature of the identified mutations, we have now included a supplementary table listing all mutations that have been mapped and characterized from the previous cited study. See **Table S4**.

8.Line 222 the same sentence is repeated here with each repetition having a different citation.
The second sentence has been removed.

9.Line 224 I found it very difficult to extract any meaning from this sentence. There's so much going on in here, you need to unpack it into a few different sentences so that you can clearly express these ideas, and provide the motivation of what you're going to do next.

We have rephrased that sentence and hope it is now clearer. **L243-248**

10.Line 233 onward. It is first pointed out that for all four traits in populations varied in the slope and the sign of the slope, but is then mentioned (in parentheses) that the only plausible explanation for this is the number of mutations in the different clones. Then this phenomenon is referred to as “stochastic radiation”. First, the different possible explanations for the observed patterns of reaction norm variation need to be properly explicated, then you need to clearly explain why the accumulation of mutations is the best explanation. The writer is making too many leaps here and expecting the reader to keep up- also it's very difficult to assess any of the claims made here.

We rephrase this section to make the point on stochastic radiation clearer. The reference to “stochastic variation in mutational input” does not refer merely to stochastic variation in the *number* of mutations (although that does vary stochastically), but also – and we think more importantly – to stochastic variation in *what* mutations occur and, in the case of identical mutations occurring in different populations, *when* they occurred. **L255-L261**

REVIEWERS' COMMENTS:

Reviewer #2 (Remarks to the Author):

I think the authors adequately addressed all the issues raised in the first round of reviews. I particularly appreciate the new Fig 1 and the new data and analyses associated with Fig S6.

Reviewer #3 (Remarks to the Author):

The authors have done a good job addressing my concerns, which were mainly about the clarity of the manuscript. I have no further concerns. Congratulations on completing a really nice study.